# Does the Calcaneus Serve as Hypomochlion within the Lower Limb by a Myofascial Connection?—A Systematic Review

**DOI:** 10.3390/life11080745

**Published:** 2021-07-26

**Authors:** Luise Weinrich, Melissa Paraskevaidis, Robert Schleip, Alison N. Agres, Serafeim Tsitsilonis

**Affiliations:** 1Center for Musculoskeletal Surgery, Charité—Universitätsmedizin Berlin, 13353 Berlin, Germany; luise.weinrich@charite.de (L.W.); melissa.paraskevaidis@charite.de (M.P.); 2Technische Universität München, 80333 München, Germany; robert.schleip@tum.de; 3Diploma University of Applied Sciences, 37242 Bad Sooden-Allendorf, Germany; 4Julius Wolff Institute, Berlin Institute of Health and Charité—Universitätsmedizin Berlin, 13353 Berlin, Germany; alison.agres@charite.de

**Keywords:** hypomochlion, Achilles tendon, plantar fascia, fulcrum, calcaneus

## Abstract

(1) Background: Clinical approaches have depicted interconnectivity between the Achilles tendon and the plantar fascia. This concept has been applied in rehabilitation, prevention, and in conservative management plans, yet potential anatomical and histological connection is not fully understood. (2) Objective: To explore the possible explanation that the calcaneus acts as a hypomochlion. (3) Methods: 2 databases (Pubmed and Livivo) were searched and studies, including those that examined the relationship of the calcaneus to the Achilles tendon and plantar fascia and its biomechanical role. The included studies highlighted either the anatomical, histological, or biomechanical aspect of the lower limb. (4) Results: Seventeen studies were included. Some studies depicted an anatomical connection that slowly declines with age. Others mention a histological similarity and continuity via the paratenon, while a few papers have brought forward mechanical reasoning. (5) Conclusion: The concept of the calcaneus acting as a fulcrum in the lower limb can partially be supported by anatomical, histological, and biomechanical concepts. Despite the plethora of research, a comprehensive understanding is yet to be investigated. Further research exploring the precise interaction is necessary.

## 1. Introduction

Heel pain represents a common pathology with a rising prevalence, especially in the elderly [1]. Approximately one in three people has experienced symptoms, including pain at the posterior or plantar part of the heel [2]. Common diagnoses are mechanically attributed pathologies such as plantar fasciitis, heel spurs, or Achilles tendinopathy [3]. With the increase of recreational and competitive sports, particularly in individuals older than 40, heel pain incidence has risen over the past decades [4]. Plantar fasciopathies and Achilles tendinopathies are common sports injuries but are most prevalent in recreational and elite runners with a lifetime incidence of almost 10% [5]. Nonetheless, non-athletes are not excluded from heel pain [6].

Evidence-based therapy for heel pain is still lacking, which, to date, increases the risk for long-term morbidity and loss of working days [6]. Interestingly, the incidence of plantar fasciitis is higher in women [7], while the incidence of pathologies of the Achilles tendon (AT), especially Achilles tendon ruptures, are higher in men [8]. Furthermore, foot pain in the elderly is strongly associated with a decreased ability to perform daily activities and a negative impact on balance and gait, leading to increased risks of falls [9]. Despite this previous research, detailed investigations of the intricate and delicate structure and function of the fascial structures involved in heel pain have not been performed.

A few studies have proposed the hypothesis of a relationship between the AT and the plantar fascia (PF), focusing on the interconnectivity of the calcaneal bone, the AT, and the PF. This hypothesis is based on a broad aspect of observations, findings, and studies [10,11,12].

Most existing studies approach the research question mainly from two angles: either from the functional side or from the anatomical side.

In the following systematic review, we analyze the studies investigating the aforementioned relationship with consideration of the following question: Is there a myofascial connection between the Achilles tendon and plantar fascia using the calcaneus as a hypomochlion? In consideration of this question, it is critical to include the anatomy of the myofascial tissue connection, as well as the biomechanical function of the calcaneus within this construct. The present review therefore reports on the anatomical (macro and micro) and biomechanical relationships of the tendinous structures of the hindfoot, as well as the distinct function of the calcaneus as a hypomochlion, in order to elucidate potential therapeutic approaches to foot and ankle pathologies.

## 2. Results

### 2.1. Biomechanical Function and Interrelations

To understand the hypothesis of the calcaneus serving as a hypomochlion in the lower limb from a biomechanical point of view, it is first important to define hypomochlia in general. A hypomochlion, or pivot point, is a sesamoid bone that serves as a point of rotation. It is described as a fixed point, or a fulcrum, in which a solid body can rotate under the action of forces. A hypomochlion increases leverage and thereby reduces the force necessary to move corresponding joints. Additionally, sesamoid bones prevent tendon damage at the joint from fluctuating pressure loads [13]. One example of a well-known hypomochlion is the patella, which is the largest in the body. It connects to the quadriceps femoris and enables full knee extension [14]. The patella is cushioned by three fat pads, yet the full extent of the connection of these fat pads with hypomochlia is still under investigation [15]. Another example is the pisiform bone, which serves the flexor carpi ulnaris and contributes significantly to the stability of the ulnar column [16].

The transfer of these concepts of a hypomochlion to the calcaneus has taken place over the past few decades. The first to present this hypothesis were Arandes and Villadot in 1953 [17]. The calcaneus can be viewed as a fixed point of the lower limb, which evolved and became more prominent with upright movement. In the evolutionary development of bipedal adaptation, the calcaneus has consequently adapted in shape and size. This includes the AT “bending” in a 90° angle around the heel, enabling bipedal walking by converting the force of the lower leg into the foot via adequate plantar flexion. This process fulfils the definition of a hypomochlion. Therefore, the posterior calcaneus must process high forces [17]. Introduced in 2014, the theoretical model by Huerta [18] explains the possible functional and pathological relationship between the gastrocnemius muscle via AT to the PF in detail [18]. Tightness of the gastrocnemius muscle leads to higher AT tension during weight bearing. This induces increased stiffness during dorsiflexion within the ankle joint, thus leading to higher hindfoot plantar flexion and an increase in forefoot plantar pressure [18]. This results in a collapsed foot arch and ergo increases the PF tension in the longitudinal plane, counteracting the arch flattening effect [18]. The model considers weight-bearing as an essential influential factor instead of a planar transmission of tension from the gastrocnemius to the PF [18].

It can be proposed that a formerly connected AT, reaching all the way from the dorsal lower leg to the foot sole, becomes divided by the calcaneus, as an ossifying bridge. This reduces the force necessary to walk [18]. In other words, based on the biomechanical interactions within the lower limb, the PF could be viewed as a continuation of the AT to the foot sole, inserting at the heads of the Ossa metatarsalia [18].

Cheung et al. constructed a 3D model of a 26-year old male using MRI images. This study exemplified the effect in the standing foot [19]. Higher load on the AT leads to increased PF tension, owing to flattening of the foot arch [20]. Another computational 3D model of a 30-year-old male’s foot further concluded that a higher AT tension during gait correlates with an increased PF stiffness [21]. This was suggested as an influential interaction of the two connective tissues [22].

In the work of Liu et al., using shear wave elastography, results indicate that different knee and ankle flexion/extension positions can influence both AT and PF. The stiffness of the AT was higher in a fully extended knee than in a 90° flexion, whereas the plantar fascia showed the opposite correlation, once again supporting a possible interaction of these tissues. In addition, in a 90° flexed knee, a significant increase in stiffness was seen within the PF and AT during ankle dorsiflexion. Interestingly, only the proximal end of the PF appeared to increase in stiffness. Generally speaking, a biomechanical connection between gastrocnemius, AT, and PF is widely accepted [23].

A work by Huang et al. investigated stiffness using a MyotonPro device and found an increase in AT and PF stiffness during higher ankle dorsiflexion, and an even higher increase in combination with knee extension [24]. In another study using a MytonPro device in a wider cohort by Orner et al., there was a positive correlation in all investigated parameters (frequency, decrement, stiffness, creep, and relaxation), further implying a functional connection of the two tissues [25].

As previously outlined, changes occurring in the fascial continuity of the lower limb during the lifespan have an embryological, ontogenic, and biomechanical causation [17]. Consequently, fascial continuity is reflected in early development, representing the functional adaption of optimal pressure distribution over a wider area. Further changes include the development of fat pads and the calcaneal of fat padding. The role of fat pads in correlation with hypomochlia is still under exploration. Currently, it is suggested that they play a role in stress minimization and joint articulation guidance, similar to the knee joint [15].

The calcaneal padding superficially surrounding the calcaneus is described as a “water-pillow” functional system. This accommodates the mechanical requirements for stability, impact resistance, and buffering caused by torsion. This is achieved by a dense division of the subcutis by flexible fibers [26]. In addition, the heel pad is secured by a strong attachment of tethering retinacula along the calcaneal tuberosity and proximal plantar fascia, preventing its dislocation during stance in gait and complementing the calcaneal function [11]. This exhibits a high level of adaptation to the function and dynamism of the calcaneal hypomochlion [11].

It can be summarized that the hypothesis of the calcaneus serving as a myofascial connected hypomochlion can also be supported biomechanically, as overarching force transmission seems to influence the fascial structures of AT and PF.

### 2.2. Anatomical Characteristics and Interactions

Anatomically, there are distinct indications that the AT-calcaneus-PF complex seems part of a broader myofascial system found within the entire body.

In the 1990s, a myofascial continuity of the dorsal human body was described by Tom Myers, who dissected human cadavers and followed anatomical soft tissue lines. The so-called “superficial back line” begins at the epicranial fascia, continues over the paravertebral muscle fascia, the sacrolumbal fascia, and the fascia of the hamstrings, then the gastrocnemius, and finally to the AT, calcaneus, and PF, ending in the fascia of the short toe flexors [27].

A collagen fascicle continuity via the superficial posterior and inferior calcaneus is apparent, especially in children and young adults, according to the investigations of ten human cadavers by Snow et al.; histological sections of fetal and neonatal feet displayed a continuous heavy layer of collagen fibers, where the AT wraps around the calcaneus and continues into the PF [11].

In a foot of a 26-year-old individual, most collagen fiber substance are inserted as superficial fibers of the tendon and fascia directly into the insertion area of the calcaneal tuberosities, eventually assuming the form of the periosteum. In the dissected feet of the elderly, this described continuity was no longer visible. Here, the bone-tendon and bone-fascia enthesis appear irregular with extensive interdigitations [11]. In addition, Snow et al. found that the paratenon of the AT continues into the superficial layer of the calcaneal fat pad [28].

Stecco et al. investigated 12 leg specimens and investigated 52 MRIs of the hindfoot with symptoms of non-specific ankle and foot pain. In the histologically investigated cases, the PF continued over the calcaneal bone with a thin band corresponding to the calcaneal periosteum [10]. Interestingly, this layer, surrounding the calcaneus was in continuity with the paratenon of the AT. The connection between AT and PF in this study was shown to take place through the paratenon [10]. The relationship between PF- and AT-paratenon was further established by the MRI findings, which implied a positive correlation between the thickness of both tissues [10].

In another study by Zwirner et al., macroscopic observations of the collagen tissue of nine human cadaver feet found a bridge at the calcaneal periosteum connecting the AT and PF [29]. At the area of the AT insertion, at the calcaneal tuberosity, there seemed to be a stronger connection of connective tissue, leaving a frayed tissue area after dissection. Furthermore, fibrous septae within the AT at the calcaneal insertion appeared to continue into the superficial trabecular meshwork of the calcaneus, differing from the rest of the calcaneal trabeculae in terms of a darker appearance, macroscopically and microscopically [29]. At the calcaneal tuberosity, finger-shaped bony extensions flowing in the trabecular architecture were detected [29]. Similar observations were made distally at the insertion of the PF. Here, the trabecular course of the calcaneus corresponded to the orientation of the collagen bundles of the PF [29]. This is in accordance to Wolff’s law, stating that bone will remodel over time to resist loading [30]. Further, decalcified samples of the posterior calcaneus were investigated using uniaxial tensile tests. By decalcification of the bone, a collagenous scaffold remained. They provided a certain structural integrity of collagenous scaffold within the bone, independent of the degree of ossification [29]. Compared to tensile strengths within the PF, it was much lower in the calcaneal and periosteal samples, though a smaller amount of collagen tissue within the calcaneal periosteal samples were identified [29].

Singh investigated 19 specimens histologically and radiologically using X-rays. The histological investigations showed at least a partial continuation of the AT fibers in the trabecular meshwork in 73% of the AT and 88% of the PF histological sections [12]. The collagen bundles inserted into the same tensile stress direction as the trabeculae. However, the radiological and histological images lacked evidence of a direct relationship between AT and PF [12]. In the analyzed micro-CT scan of six specimens, a dense trabecular system in the posterior calcaneus was identified [12]. The paratenon continues into the calcaneal periosteum to finally merge with the PF, providing further evidence of a soft tissue connection [12]. In general, strong correlations were seen within the cross-sectional areas. As expected, the thickness of the AT at its insertional area and the PF’s insertional length correlated directly, as did the cross-sectional areas of the insertional lengths of both AT and PF [12].

Furthermore, Milz et al. reconstructed a 3D model of the AT insertion of four cadavers using histological analysis and radiological imaging. They were able to distinguish the calcaneal trabeculae alignment along the direction of the AT fascicles and orientation towards the proximal attachment of the PF [31].

Another study by Kim et al. investigated 60 human cadaveric limbs of different age groups. They found that about 8% had a lower calcaneal insertion of the Achilles tendon and retained a connection between AT and PF [32]. The cases displaying this connection were the youngest of the investigated ones, between 43 to 48 years old. None of the specimens showed a complete continuity of the fibers from AT and PF [32]. In an MRI study conducted by the same research team, the AT insertion in the elderly showed a more proximal area of insertion, compared to younger people [32]. Together, the AT’s insertion appears to migrate proximally by 0.63% per year. In two young subjects, 12 and 16 years of age, a continuous relationship between AT and PF was reported [32]. Interestingly, a study by Pekala et al. showed that the insertion of the PF does not change in comparison to the changing insertion location of the AT [33].

In a study by Ballal et al., 12 fresh frozen specimens and 10 embalmed specimens were dissected to investigate the continuity of the AT into the PF. In only three of the specimens, the fascicles of the tendon of the medial gastrocnemius insertion, the most distal area of insertion into the posterior calcaneus, were continuous with the PF through the periosteum [34]. Again, these findings were mainly seen in younger cadavers [34].

First described in 1953, a model of the biomechanical connection of the Achilles tendon-calcaneus-plantar fascia system was introduced by Arandes and Viladot, as already touched upon previously [17,35]. The initial hypothesis was that the calcaneus, as a big sesamoid-like bone similar to the patella, transmits force of the AT through the posterior trabecular system of the bone into the PF and short flexors of the foot [17]. This idea was based upon an ontogenic view. The ontogenic point of view of the hypothesis has been researched by carrying out histological investigations. The histological investigations of the embryogenic cadavers showed early fibrous connections of AT and PF through the calcaneus [17]. Arandes und Villadot also described a fading of the connection around the age of 7 [17].

Newer observations of anatomic development in fetuses describe continuous fibers between AT and PF, merging into the thickened perichondrium in the posterior part of the main calcaneal body [36]. Shaw describes the enthesis organ as an enthesis fibrocartilage and a sesamoid fibrocartilage in the deep surface of the tendon as well as a corresponding periosteal fibrocartilage covering the superior calcaneal tuberosity [36]. It is stated that both the developing AT and PF initially attached to the perichondrium of the calcaneus [36]. Additionally, a heel fat pad appears to be connected to the AT and calcaneus in the superficial layer [20,37].

This leads to another structural argument, which is found in the anatomical features of hypomochlia, as mentioned earlier: the fat pad cushioning. In the knee, the infrapatellar fat pad is part of its enthesis organ [37]. As a classic example of a hypomochlion, the patella is cushioned by the anterior and posterior suprapatellar and the infrapatellar fat pad [15]. The fat pad of the AT-calcaneus-PF complex is found in the AT enthesis organ [36]. The enthesis organ consists of a sesamoid fibrocartilage in the tendon’s deep surface, adjacent to its attachment site, and a complementary periosteal fibrocartilage on the posterior calcaneal surface [20,37]. The retrocalcaneal bursa, which serves as the distal outer layer of the retromalleolar fat pad, or Kager’s fat pad, is in anatomical contact with those surfaces [38,39]. In addition, the calcaneus has a calcaneal padding. While the patellar fat pads are separate pads lying in close proximity to the joint and do not serve as cushions towards the skin, the calcaneal cushioning encircles the whole bone. A detailed anatomical work describing the anatomy of the heel’s connective tissue was conducted by Erich Blechschmidt [26]. The water-pillow effect of the calcaneal padding was described earlier. [26]. This is achieved by a dense division of the subcutis by flexible fibers [26]. In addition, the heel pad is secured by a strong attachment of tethering retinacula along the calcaneal tuberosity and proximal plantar fascia, preventing its dislocation [11]. This fat pad cushioning is another additional feature unique to the calcaneal hypomochlion.

In light of the collective findings of the studies reviewed above, it is strongly supported that a fascial continuation exists between AT and PF, including the calcaneus. Studies that did not find this continuous connection nonetheless described its existence in younger specimens. This is most probably attributed to the fact that the AT-calcaneus-PF connection is dynamic and changes over the years. As the AT seems to migrate more proximally over time, the connection within such tissues becomes reduced, but a thin continuation seems to persist.

### 2.3. Histological Structure

Another element to consider in the hypothesis of a single tendon origin is the immunohistochemical composition of the AT and PF. Both tissues have tenocytes and tenoblasts as the main cellular elements, as well as collagen I and III fibers as the main elements of the extracellular matrix. Tenocytes and tenoblasts comprise up to 90% of the cellular elements in the AT. The remaining 10% comprise chondrocytes, vascular cells, and smooth muscle cells. Within the PF, chondrocytes found at the proximal end of the tissue are an expression of possible cartilaginous metaplasia [10,40].

Collagen constitutes the largest amount of the dry mass of healthy tissue in the two tissues, being about 60–85% of a healthy tendon [10,40,41]. Apart from non-collagenous glycoproteins, small amounts of proteoglycans are included in the composition, influencing formation and regulation of hierarchical collagen construction [41,42]. Type III collagen is another major variant found within the tendon and the fibrocartilage material [43]. In lesser quantities, Type II and V collagen are present, but not within the midportion of the tendon. Type II collagen is principally located in the fibrocartilage region. Type V collagen forms the core of Type I collagen [43]. These elements demonstrate a capacity to build dense fibrous connective tissue, as well as hyaline cartilage [22]. Within the PF, the extracellular matrix is mainly formed by collagen and elastin. Immunohistochemical analysis has found that almost all PF tissue is made up of Type I collagen. Type III collagen only appears in the loose connective tissue and the large, multidirectional fibrous bundles. In lesser quantities, type II collagen is included in both fascial tissues, but it is mainly found towards the heel, the fibrocartilage region of the AT, and around chondrocytes in the PF [10,43]. Loose connective tissue also appears to include thin elastic fibers, as well as glycosaminoglycans or hyaluronan [10]. This allows the plantar fibrous bundles to glide towards each other and absorb compression [44].

The AT collagen fibers form fascicles as very tightly packed, parallel bundles. The fascicles are surrounded by endotenon, and the entire tendon is encoded by epitenon. In turn, this is covered by paratenon, forming the macroscopic tendon [10]. The function of those two “tendons” (epi- and paratenon) is thought to reduce friction and allows for smooth motion [45]. The typical parallel, tightly packed order of the AT fascicles loosens slightly in the PF. The proximal to distal direction is constant, but some loose connective tissue is added [10].

Generally, a lower quantity of type II collagen is found in both fascial tissues, primarily towards the heel, specifically in the fibrocartilage region of the AT and around chondrocytes in the PF [10,43]. Chondrocytes towards the proximal end of the PF are considered to be the expression of possible cartilaginous metaplasia. This could explain the shrinking connection between AT and PF in the elderly [32]. Its purpose could be an adaption to the high mechanical loads in this area [22].

Interestingly, the PF in particular seems to be well-innervated with numerous nerve endings and Ruffini and Pacini corpuscles [10]. These nerve endings are particularly abundant at the joint point of the PF abductor hallucis and abductor digiti minimi muscle fascia and where the flexor muscles insert. These abundant innervations suggest that the PF plays an important role in proprioception, aiding stability and control of foot movements [10,46].

In summary, the similarity in the cellular and extracellular elements found within AT and PF suggest and support an original link of the two.

## 3. Discussion

This paper applied the methodology of a systematic review to investigate the anatomical and functional connectivity of the AT-calcaneus-PF system and its implications on the hypothesis of the calcaneus as a hypomochlion of the fascial structures. Each included study assessed a different aspect of the AT-calcaneus-PF system. Read in conjunction with the hypothesis of a physical-anatomical and mechanical entity, the individual findings support our hypothesis.

This work underlines the low number of included articles, which highlights the potential of the research question for investigation in further research.

All studies assessing the anatomical connection reported homogenous findings. This anatomical continuity, however, was not always overtly obvious in cadaveric properties. Nonetheless a fiber continuity from the paratenon onto the superficial layer of the calcaneal periosteum into the PF was detectable. This is most probably attributed to the fact that the connection is dynamic and remodels over the lifespan. The continuity is reduced over time as the insertion of the AT migrates proximally. Additionally, the histological findings displayed a high degree of uniformity in the collagenous makeup and reinforced the common ontogenic origin. While numerous studies approached the question of continuation of AT into the PF from an anatomical point of view, few studies compared the cellular and extracellular components of each tissue. An analysis on this microanatomic level might bring further insight into the research question discussed in this paper.

While the understanding and research on a macro- and microscopic and anatomic level is cardinal, a lack of exhaustive understanding is not necessarily a hindrance for effective clinical practice. While embryological origins and links are still speculated upon, pathologic changes in either the AT or PF have long been known as aetiologies for different pathologies.

Pathological degeneration of the AT can cause plantar fasciitis. A widely accepted, conservative approach includes stretching exercises. The application of these is based on the premise of the mechanical and anatomical link between the two structures. Calf stretching, night splints, and high-loading strength training targets tightness and stiffness of the AT. Another aspect not extensively addressed is the evolutionary point of view. Anthropologists have proposed that the ability of improved running of humans leads to their typical leg anatomy [47]. Lower limb anatomy appears to be influenced more by the demands of running rather than walking or climbing. In particular, during running, the lengthening capacity of the AT in humans can reduce muscular demand [48], and a longer AT positively impacts energy efficiency. Prior to the use of cushioned shoes, running was mainly performed by landing on the forefoot. During the swing phase of forefoot running, the calcaneus can be considered as a hypomochlion in the flexible element of the leg. This hypomochlion would then lengthen the tensile stress of the AT in the PF and could be viewed as an indirect indication of the connection between the two fascial structures through the calcaneus.

Further, this could be supported by the functional and biomechanical interplay affirmed by the windlass mechanism. Dorsiflexion of the toes increases the strain on the PF. Adding force to the AT further strains the PF. Thus, the whole PF is under an increased load, causing a stress concentration near the medial calcaneal tubercle and the first foot ray, decreasing gradually towards the lateral rays of the foot. Once again, this implies a ubiquitous mechanical and functional interdependence [49].

Considering the effectivity of these management options, these collectively imply a fibrous continuation in the mediolateral axis of the AT and PF.

## 4. Materials and Methods

The structure and content of this systematic review is based on the SPIRIT checklist, and the search strategy was developed using the PRISMA guidelines.

To identify studies for the systematic review of our research question, two authors independently screened all titles and abstracts. In case of differing opinions on inclusion or exclusion, the studies were analyzed together in detail and discussed with a third author. The databases Pubmed and Livivo were used as a comprehensive database in this case, including the databases MEDLINE, ZB MED, BASE, and Thieme Publisher’s Database. The full-text papers of any published study considered to be relevant were obtained and read. Each obtained study was assessed and audited based on the inclusion criteria, as defined in Table 1. No limitations or restrictions were made with regards to the date of publication, journal type, or language of publication. Nonetheless, most studies were in English. We were fortunate to receive help with the translation of Spanish studies. All other studies, duplicates, and off-topic studies were excluded. Off-topic studies, for example, were clinical trials on surgical techniques, other therapeutic possibilities for heel pain or spurs, or studies on ankyloses spondylitis. The search terms used were “Achilles tendon”, “plantar fascia”, and “calcaneus”. For the Pubmed search, they were combined as follows: ”Achilles tendon” AND “calcaneus” OR “Achilles tendon” AND “plantar fascia” OR “calcaneus” AND “plantar fascia” as well as a search combination of “Achilles tendon” and “calcaneus” and “plantar fascia”; for Livovo, these terms were used without any additional Boolean operators. The last databank search was conducted in May 2021.

The study flow is depicted in Figure 1. The search in two databases yielded a total of 328 records. After removing duplicates (*n* = 57) and excluding articles not addressing the research question (*n* = 252), a total of 19 studies were included that addressed the research question: Does the calcaneus serve as a hypomochlion within the dorsal lower limb by a myofascial connection?

An overview briefly summarizing the nineteen included studies can be found in Table 2. A comprehensive list of the included studies can be found as an Appendix A to this paper.

## 5. Conclusions

In summary, according to the modest number of studies addressing the research question, the majority are in favor of the hypothesis of an anatomical connection between the AT and PF through the calcaneus. However, the disparity of fiber continuation in young and elderly cadavers is subject to critical reassessment. In the younger cadavers, a direct anatomic connection was evident, which provides strong support for the imminent biomechanical interaction. Persistent biomechanical and stressors during the stance phase and gait phase can affect the AT and PF differently, but biomechanical interdependence of the tissues remains. Conceivably, this can lead to an imbalance and shortcoming of equilibrium, inducing a number of possible foot pathologies. From a biomechanical point of view, both load and tension on the AT influence PF stiffness during standing, walking, and other activities that engage a range of different knee and ankle angles.

Combining anatomical, biomechanical and histological observations, the tendinous connection in the elderly seems to be less prominent, or even partially interrupted. From a biomechanical approach, a functional connection influences forces on either side of the calcaneus. In later stages of the lifespan, the former tendinous connection seems to gradually be separated by ossification, as the posterior trabecular meshwork of the calcaneus shows an anatomically different configuration compared to the rest of the bone. The trabeculae appear in the same tensile direction as the former singular tendon, separating the AT and PF by a soft tissue and trabecular bridge.

Considering collagen fibers in the calcaneal trabecular system, such a connection would also exist in the elderly. From a clinical viewpoint, the fascial connection is already used by therapists in rehabilitation treatments for pathologies of AT and PF, although a detailed understanding of the therapeutic influence remains unclear. While some of the mentioned aspects and compelling arguments advance the hypothesis of a physical-anatomical and mechanical unit, further studies must be performed in order to further explore and confirm this concept.

## 6. Limitations

While this review highlighted, investigated, and analyzed the relationship between the AT and the PF and the calcaneus, limitations in this study should not be overlooked.

Application of the inclusion and exclusion criteria to the results of the databank searches identified 19 studies for this review from an initial search yielding 328 studies. Limited research has been performed with regards to this topic. Compared to other topics, this is a surprisingly small number of applicable and relevant studies, given the epidemiology of lower limb pathologies. The search in different databases, along with supplementing and expanding the search by hand and cross-researching lists of reference, allow confidence in the inclusion of relevant research.

While we predominantly included studies conducted in English, we were able to include a Spanish-language paper with the aid of a native speaker translation.

Further, the methodological search strategy using the three main keywords was appropriate to narrow down the search results, but may have excluded studies that examined each entity separately or that focused on an isolated aspect in great detail, irrespective of other structures and their interconnectivity. Further, the nature of the hypothesis draws on multifaceted aspects ranging from embryological and histological to macroanatomic and mechanical aspects, making it difficult to find studies addressing all or some of those aspects. An important aspect to consider is the heterogeneity of the type of studies included. The study type ranged from cadaver studies to analyses of MRI studies.

Additionally, an inevitable limitation is posed by the chosen databases and database access provided by our institution.

The nature of the hypothesis generally excludes the possibility of conducting in vivo and prospective studies; thus, most included studies investigated cadavers.

Future research should consider and address the conceptual limitations of the current findings. These studies should concentrate on the relationship between histological and anatomical relationships, as well as clinical ramifications and transferability.

## Figures and Tables

**Figure 1 life-11-00745-f001:**
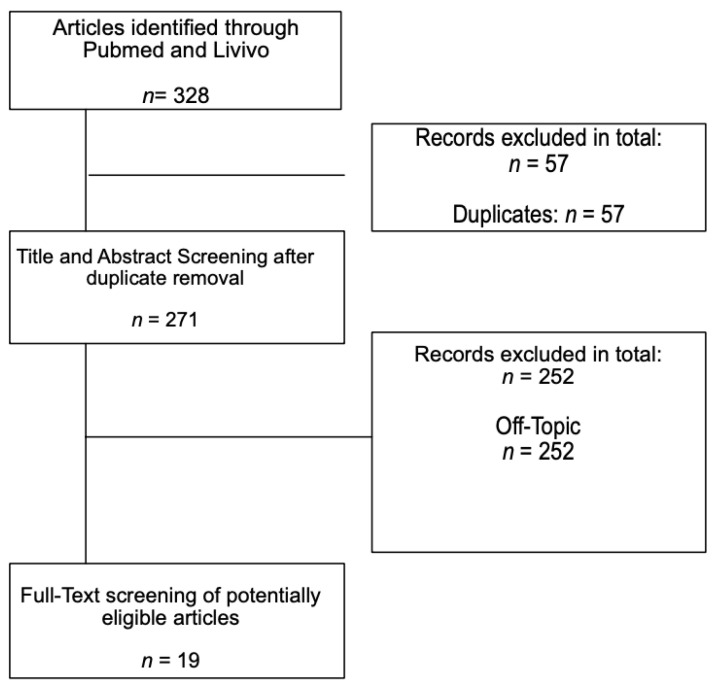
Flowchart of the included studies.

**Table 1 life-11-00745-t001:** Inclusion and exclusion criteria for the studies included in the systematic review.

Inclusion Criteria		Exclusion Criteria
**Study Setting**	Any,	Foreign languages other than German, English, and Spanish;
	no date restriction	
	no journal restrictions,	Digital unavailability
	no language restrictions	
**Study design**	Any, except for other systematic reviews	Systematic review;
		Reviews
**Study aim**	Necessary to analyze or examine the interplay between the three structures “Achilles tendon”, “calcaneus”, and “plantar fascia” or the interplay between at least two structures	Assessing current knowledge;
		No analysis of interconnectivity of the named three structures

**Table 2 life-11-00745-t002:** Categorization of included studies focusing on the interconnectedness of the Achilles tendon and plantar fascia from an anatomical, histological, or biomechanical point of research.

Research Approach	Study	Sample Size	Focus of Examination	Conclusion Concerning Connection of AT and PF
**Anatomical combined Functional**	Arandes and Viladot, 1953	-	Ontogenic (histological) and phylogenic anatomic (animal) observations.	Theoretical model: AS-calcaneus-PF system with anatomic and functional continuity.
**Functional**	Cheung et al., 2006	6 feet of middle-aged male cadavers	3D finite element model of human foot and ankle using fresh cadaveric ankle–foot specimens investigating loading response in the standing foot of AT and PF.	Increase in AT load leads to PF increase in PF tension (by arch flattening).
	Shaw et al., 2008	17 feet of spontaneously aborted fetuses, 7–32 weeks and of 6 adults	Histological sections of fetuses’ feet and MRI of adult feet.	Continuing fibers between AT and PF as a thickened perichondrium in the posterior part of the main calcaneal body.
	Chen et al., 2014	1 foot of a 30-year-old cadaver	3D finite element model of a human foot and ankle was developed using computed tomography images.	Higher AT force during gait correlate with higher PF tension during gait.
	Huerta et al., 2014	-	Theoretical model of force transmission within gastrocnemius, AT, calcaneus, and PF.	Gastrocnemius tightness leads to higher AT tension, finally increasing higher plantar tension and stiffness of the PF.
	Huang et al., 2018	15 female and 15 male healthy participants, average are 22.9 ± 3.8 years	Correlation of frequency, decrement, stiffness, creep, and relaxation between AT and PF using Myoton Pro device.	Increase in AT and PF stiffness during higher ankle dorsiflexion, even higher during knee Extension.
	Orner et al., 2018	150 healthy males and females, mean age 33 years	Correlation of frequency, decrement, stiffness, creep, and relaxation between AT and PF using Myoton Pro device.	Significant positive correlation of AT and PF.
	Liu et al., 2020	20 males, 19–23 years	Passive elastic properties of triceps surae, AT and PF measurement with shear wave elastography in different knee and ankle angles.	During ankle dorsiflexion, AT and PF show an increase in stiffness, while 90° knee flexion and stiffness of AT and PF seem to be correlated oppositely according to knee flexion and extension.
**Anatomical and histological**	Snow et al., 1995	15 feet from human cadavers	Histologic dissection and observation of human cadaveric feet.	In fetuses, there is a continuous collagen layer within the AT wrapping around the calcaneus into the PF. In older feet, the continuity is not visible anymore.
	Milz et al., 2002	4 feet of male and female cadavers, 32 to 73 years old	3D model of the AT insertion of 4 cadavers.	Alignment of the calcaneal trabeculae along the direction of the AT fascicles in orientation towards the proximal attachment of the PF.
	Shaw et al., 2008	Fetuses of 17 different crown rump lengths	Histologic investigation of sectioned fetuses.	AT and PF merging into the thickened perichondrium in the posterior part of 152 the main calcaneal body, thus AT and PF initially attached to the perichondrium of the calcaneus.
	Myers, 2009	-	Anatomical investigation as macroscopic preparation of human cadavers.	Superficial back line flowing into anatomical connection of AT and PF.
	Kim et al., 2010	60 human cadaveric limbs, of 40 cadavers, 43 to 98 years old	Macroscopic visualization and palpation of cadaveric feet.	Eight percent had a lower calcaneal insertion of the Achilles tendon and retained a connection between AT and PF in younger specimen (43 and 48 years old). None of the specimen showed a complete continuation.
	Kim et al., 2011	69 MRIs, 10 to 40 years old	MRI review from database by radiology group.	In two young subjects, 12 and 16 years of age, a contiguous relationship between AT and PF was seen. AT’s insertion location seems to migrate proximally by 0.63% each year of life.
	Stecco et al., 2013	12 feet from unembalmed human cadavers, 67 to 92 years old, and 52 MRIs, mean age 44.2 years	Histologic dissection and observation of human cadaveric feet, also microscopically, and MRI investigation of patients with heel pain.	The paratenon of the AT is in continuity with the periosteum of the calcaneus and that is in continuity with part of the PF. Addoitionally, cellular and extracellular components seem similar.
	Ballal et al., 2014	12 fresh frozen specimens and 10 embalmed specimens, 57 to 95 years old	Macroscopic visualization of cadaveric feet.	Only three of the specimens showed a continuation of AT fascicles with the PF through the periosteum.
	Pekala et al., 2019	202 MRIs	MRI review from two observers.	Insertion of the PF does not change in comparison to the changing insertion location of the AT.
	Zwirner et al., 2020	9 feet of human cadavers, 28 to 93 years old	Macroscopic and histologic investigations.	At the calcaneal AT and PF insertion, the fibers from both sides seem to continue into the posterior calcaneal trabecular bone meshwork, appearing in the fiber tension direction.
	Singh et al., 2021	18 male and female specimen, 38 to 94 years old	Histological and radiological investigation of cadaveric feet.	Partial continuation of AT fibers in the trabecular meshwork in stress direction of the AT. AT paratenon continues into the calcaneal periosteum, merging with the PF. AT thickness has a positive correlation with the insertional length of the AT.

## Data Availability

Not applicable.

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
