# Peer review of "Does the Calcaneus Serve as Hypomochlion within the Lower Limb by a Myofascial Connection?—A Systematic Review"

_life, 2021, doi:10.3390/life11080745_

Round 1
Reviewer 1 Report
The paper is interesting. Some remarks were presented in the attached file.

Author Response
1. Why definition of the hypomochlion was only discussed in 2.3 paragraph? It could be placed at
the very beginning of the paper and used to define the discussed problem.
- Thank you. We adjusted the structure of the results.
2. It is not clear if the main question discussed in the paper is: “Does the calcaneus serve as
hypomochlion within the lower limb” or “Are the Achilles tendon and the plantar fascia
connected to each other?” It could be interesting if Authors clearly state if both these aspects
should be taken into consideration together.
- We appreciate you pointing this out. With the lines 59 – 62 we added a clarifying
sentence to highlight the importance of both questions.
3. It is not clear why Authors use “dorsal lower limb” instead of simply “lower limb”.
- We agree with the anatomical accuracy and deleted the word “dorsal”
4. Some arguments used to indicate on the possible connection of the Achilles tendon and the
plantar fascia via the calcaneus are discussed without proper understanding of the biomechanics.
It is obvious that:
a. “The trabecular course of the calcaneus corresponded with the orientation of the collagen
bundles of the PF” (lines 94-95), because it’s the routine effect of bone tissue adaptation to
loadings acting on the trabecular structure (Wolff’s law)
- Thank you for this advice. This well known and widely accepted biomechanical
phenomenon is to be seen as a supportive argument for the connection of both
fascial structures. We added a sentence for clarification (183-184).
b. The dimensions (cross-sectional area) of the Achilles tendon and plantar fascia (lines 110-
112) are mainly based on strength criteria. If the forces carried by these anatomical
structures are similar, the dimensions should also be similar.
- Thank you for this advice. Since this was one of the results stated in the cadaveric
study, it ought to be mentioned. See 199
5. Is the best solution to present “4. Materials and Methods” after “2. Results” and “3. Discussion”?
- Concerning this, we are very open to the order of the chapters, the structure we
utilized was the one layed down in the template. If another order is of
preference and we misunderstood the template, we are happy to change the
structure.
-
6. The Achilles tendon should be written with capital letter.
- Thank you for this annotation. We corrected the text.
7. What was the role of Robert Schleip and Alison Agres as the author (lines 419-426)?
- Specificed at the according point

Reviewer 2 Report
Thanks for the opportunity to review this paper.
This is an interesting study. However, I have several comments that I hope the authors will find helpful.
My main concern is regarding the methodology. Authors should review SPIRIT checklist and flow diagram (https://www.equator-network.org/). Both of them should be included as part of the manuscript.
- Author should include the complete search strategy, not only the terms used (“achilles tendon”, “plantar fascia” “calcaneus”), Any language or publication year restriction?
- More information about databases. Why did you use just two? Why Livovo (Is not quite usual in other countries)
- Information about inclusion and exclusion criteria is limited. The complete information should be included in the flow diagram and/or text
- Considering the number of references (n=183) maybe authors should consider other database or “grey literature”
- “To identify studies for the systematic review of our research question two reviewers in-348 dependently screened all titles and abstracts.”. If the two reviewers didn’t agree how it was resolve?
- Conclusion: considering the limitations I’m not sure if there is a robust evidence (a few number of article, small sample size, different type (MRI, embalmed specimens…). Authors should consider the limitations to evaluate it.
Author Response
My main concern is regarding the methodology. Authors should review SPIRIT checklist and flow
diagram (https://www.equatornetwork. org/). Both of them should be included as part of the
manuscript.
- See 368-369
Author should include the complete search strategy, not only the terms used (“achilles tendon”,
“plantar fascia” “calcaneus”)
- See 373-374 and 383-385
Any language or publication year restriction?
- See 377-378
More information about databases. Why did you use just two? Why Livovo (Is not quite usual in
other countries).
- See 373-374, Livivo is very useful as it is comprehensive database and operator
which allows for a simultaneous search of numerous databases. This database
was set up by the German Federal Ministry of Education and Research (BMBF).
Information about inclusion and exclusion criteria is limited. The complete information should be
included in the flow diagram and/or text.
- See 378-382
Considering the number of references (n=183) maybe authors should consider other database or
“grey literature”
- We agree and recognize the fact that a comparably small number of references
met our inclusion criteria and discussed this in the limitations. We assume this to
be due to the fact that the topic is not well researched yet, is highly specific and
needs more attention, which we are trying to support with this paper.
- See 327-328, 435 - 439
“To identify studies for the systematic review of our research question two reviewers independently
screened all titles and abstracts.”. If the two reviewers didn’t agree how it was resolve?
- See 373-375
Conclusion: considering the limitations I’m not sure if there is a robust evidence (a few number of
article, small sample size, different type (MRI, embalmed specimens…). Authors should consider the
limitations to evaluate it.
- That is another important point and we integrated it, see 403-405